# Morphology and Crystal-Plane Effects of Fe/W-CeO$_2$ for Selective Catalytic Reduction of NO with NH$_3$

**Feihu Liu** [1],[†] (D)**, Zhong Wang** [2],[*],[†] (D)**, Da Wang** [2]**, Dan Chen** [3] (D)**, Fushan Chen** [1],[*] **and Xuebing Li** [2],[*]

[1] College of Chemistry and Molecular Engineering, Shandong Provincial Key Laboratory of Biochemical Engineering, Qingdao University of Science and Technology, Qingdao 266000, China; Feihu_Liu1123@163.com
[2] Key Laboratory of Biofuels, Qingdao Institute of Bioenergy and Bioprocess Technology, Chinese Academy of Sciences, Qingdao 266101, China; wang_da@qibebt.ac.cn
[3] College of Environmental Science and Engineering, Yangzhou University, Yangzhou 225127, China; chendan@yzu.edu.cn
[*] Correspondence: wangzhong@qibebt.ac.cn (Z.W.); chen-fushan@263.net (F.C.); lixb@qibebt.ac.cn (X.L.); Tel.: +86-1586-551-6166 (F.C.); +86-1586-687-0310 (Z.W. & X.L.)
[†] These authors contributed equally to this work.

**Abstract:** The CeO$_2$ ordinary amorphous, nanopolyhedrons, nanorods, and nanocubes were prefabricated by the hydrothermal method, and employed as carriers of Fe/W–CeO$_2$ catalysts to selectively catalyze the reduction of NO with ammonia. Characterization results indicated that the morphology of CeO$_2$ support originated from selectively exposing different crystal surfaces, which has a significant effect on oxygen vacancies, acid sites and the dispersion of Fe$_2$O$_3$. The CeO$_2$ nanopolyhedrons catalyst (Fe/W–CeO$_2$–P) showed most oxygen vacancies, the largest the quantity of acid sites, the largest BET (Brunauer-Emmett-Teller) surface area and the best dispersion of Fe$_2$O$_3$, which was associated with predominately exposing CeO$_2$ (111) planes. Consequently, the Fe/W–CeO$_2$–P catalyst has the highest NO conversion rate in the temperature range of 100–325 °C among the ordinary amorphous, nanorods, and nanocubes Fe/W–CeO$_2$ catalysts.

**Keywords:** morphology; Fe/W–CeO$_2$; nanopolyhedra; CeO$_2$ (111) planes

## 1. Introduction

As one of the main atmospheric pollutants, nitrogen oxides (NO$_x$) can cause acid rain and photochemical smog, which are pernicious to the environment and human health [1–3]. At present, the most effective method for treating stationary sources nitrogen oxides is mainly the selective catalytic reduction of nitrogen oxides with NH$_3$ (NH$_3$-SCR) [4,5]. Different kinds of inorganic metal oxides, such as MnO$_x$, CeO$_2$, and Fe$_2$O$_3$, cause great influence because of their high activity and high selectivity for NO$_x$ reduction [6,7].

So far, more and more interest has been made in the study of the inorganic metals facet-dependent catalytic activities and inorganic metal oxide nanocatalysts, as it is known that the arrangement of atoms on the surface is directly correlation with catalytic reactivity. Inorganic metal oxide nanocrystals containing Co$_3$O$_4$ (110) nanorods [8], CeO$_2$ (110) sheets/CeO$_2$ (110)/(100) nanorods [9–11], TiO$_2$ (001) nanobelts [12–14], and Cu$_2$O (111) polyhedron/Cu$_2$O (110) rhombic dodecahedra [15,16] have also been found to show great superiority in catalytic activity in different reactions, because of the priority exposure of reactive faces in catalytic surfaces. These catalysts are exposed to a certain reaction surface and provide a reference for establishing a practical model for the basic research of heterogeneous catalysis, which are comparable to single crystal catalysts.

In addition to loading precious metal catalysts, ceria, the well-known functional rare-earth metal material, has also been used in $NH_3$-SCR. This is because its high oxygen storage capacity is related to the oxygen-rich vacancies and the low redox potential between $Ce^{4+}$ and $Ce^{3+}$. Theoretical and experimental studies have shown that the dense $CeO_2$ (111) face is the most stable (thus inherently less reactive); $CeO_2$ (110) and (100) has lower stability [17]. Therefore, understanding the relationship between the structure and relative stability of $CeO_2$ faces can provide profound insights into catalytic reactions that typically occur on the surface. It is currently believed that the $NH_3$-SCR in ceria-based catalysts has been greatly enhanced, and a large number of studies have investigated catalytically active sits, oxygen vacancies, and metal-support interactions [18].

Therefore, the major purpose of the present study was to evaluate the relationship between the structural feature and activity of Fe/W–$CeO_2$ catalysts with different morphology. We prefabricated $CeO_2$ ordinary, nanopolyhedra, nanorods, and nanocubes with various exposed crystallographic facets by hydrothermal methods. Then the resulting powders were characterized by X-ray diffraction (XRD), transmission electron microscopy (TEM), Raman spectroscopy, diffuse reflectance UV–VIS spectroscopy (UV–VIS), Pyridine Fourier Transform Infrared Spectroscopy (Py-IR) and X-ray photoelectron spectroscopy (XPS).

## 2. Results

### 2.1. TEM and HRTEM Analysis

The morphology and microstructures of the $CeO_2$ and Fe/W–$CeO_2$ were conducted by TEM and HRTEM (high-resolution transmission electron microscopy) in Figure 1. It was found that $CeO_2$ catalyst of A, B, C, D showed ordinary morphology (A1), nanopolyhedrons (B1), nanorods (C1) and nanocubes (D1) in morphology, which denoted as $CeO_2$–O, $CeO_2$–P, $CeO_2$–R and $CeO_2$–C, respectively [11]. In particular, the morphology of $CeO_2$–O includes polyhedrons and ellipsoids. After impregnating Fe and W over the $CeO_2$ carrier, no significant change of the morphology was noticed, so the Fe/W-$CeO_2$ catalysts with different morphology were prepared. The size of $CeO_2$–O, $CeO_2$–P, and $CeO_2$–C were about 10 nm, 8 nm, and 40 nm. Additionally, $CeO_2$–R showed shape with 27–150 nm in length and 5–10 nm in width.

The HRTEM image in Figure 1 shows the lattice spacings of 0.31, 0.27, and 0.19 nm, matching well with the (111), (110) and (100) planes of $CeO_2$ for Fe/W–$CeO_2$–P, Fe/W–$CeO_2$–R, and Fe/W–$CeO_2$–C catalysts, respectively [11,19]. For Fe/W–$CeO_2$–O catalyst, it exposed three crystalline planes including $CeO_2$(111), $CeO_2$(100) and $CeO_2$(111). These observations indicated that the different preparation of $CeO_2$ support could expose different $CeO_2$ crystal plane [20]. Furthermore, all Fe/W–$CeO_2$ catalysts can be observed with the well-solved periodic lattice fringe of 0.25 nm consisting of the interplanar distance of the $\alpha$-$Fe_2O_3$ (110) plane. As reported by Prieto-Centurion and Eaton et al. [21], similar $\alpha$-$Fe_2O_3$ (110) structures on a $CeO_2$ support were also presented. Especially, all Fe/W–$CeO_2$ catalysts were measured lattice spacing of 0.29 nm to index to the $FeWO_4$ (111) crystal plane. Qu [22] et al. also proposed that the $FeWO_4$ species was formed in $Fe_{1-x}W_xO_\delta$ catalysts, which could provide Brønsted acid sites on account of the formation $NH^{4+}$ as highly active species. These can express that different $CeO_2$ crystal faces did not affect the exposure of iron and tungsten crystal faces.

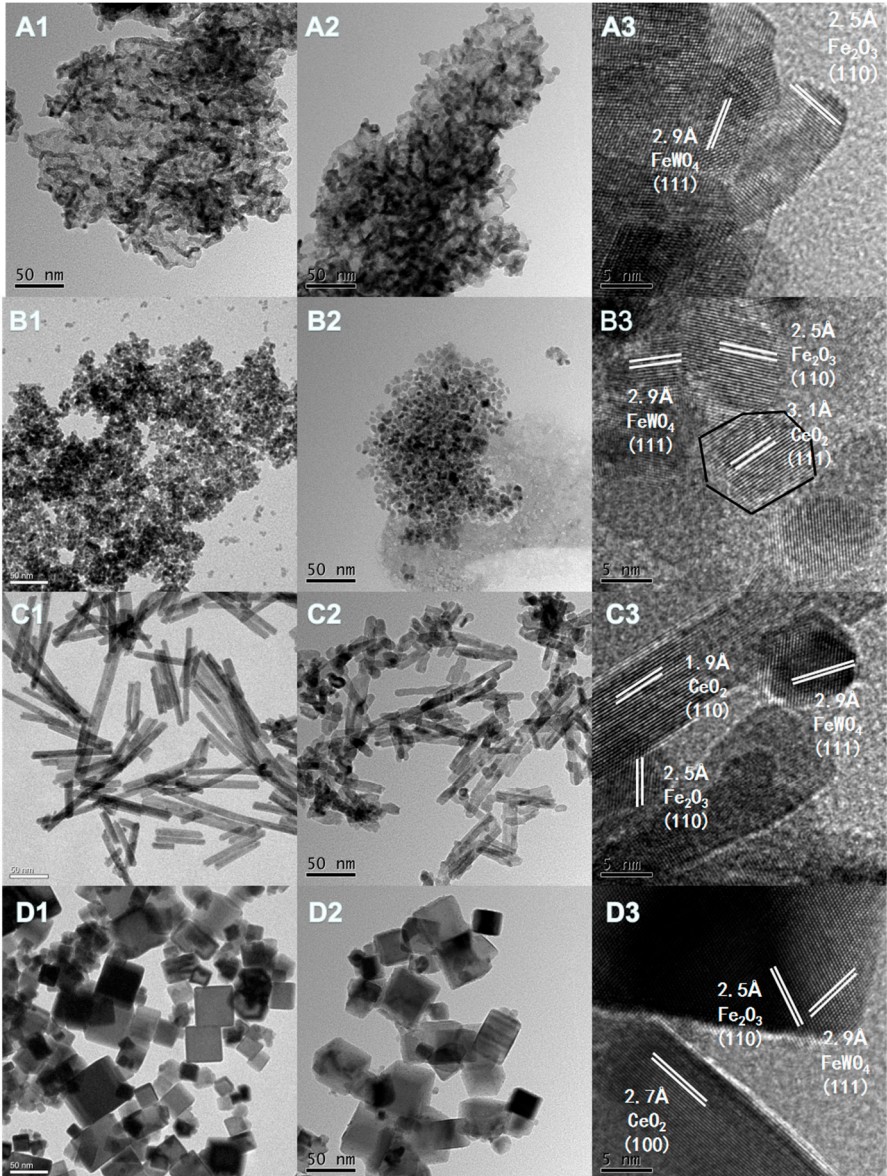

**Figure 1.** TEM images of (**A1**, **B1**, **C1** and **D1**) of CeO$_2$–O, CeO$_2$–P, CeO$_2$–R and CeO$_2$–C; TEM and HRTEM images of Fe/W–CeO$_2$–O (**A2** and **A3**), Fe/W–CeO$_2$–P (**B2** and **B3**), Fe/W–CeO$_2$–R (**C2** and **C3**) and Fe/W–CeO$_2$–C (**D2** and **D3**).

*2.2. XRD*

As shown in Figure 2, only the typical diffraction peaks of fluorite cubic structure were observed in as-synthesized materials. The diffraction peaks at 28.6°, 33.0°, 47.5°, 56.4°, 59.1°, 69.4°, 76.9°, and 79.4° can be attributed to (111), (200), (220), (311), (222), (400), (331), and (420) of the facets of CeO$_2$ (JCPDS card no. 43-1002), respectively. The wider reflection peak of the Fe/W–CeO$_2$–P indicated a small crystallite size compared with that of Fe/W–CeO$_2$–C, Fe/W–CeO$_2$–R and Fe/W–CeO$_2$–O catalysts [23,24]. The average crystalline sizes of Fe/W–CeO$_2$ were calculated from the CeO$_2$ (111) plane using Scherrer's formula. The results (Table 1) indicated that the order of crystallite size was similar to that of result of TEM. In the XRD measurement, the (011), (110), and (111) peaks of FeWO$_4$ should be at 2θ = about 23.7°, 24.9°, and 30.1°, respectively [25]. From the XRD comparison figure of CeO$_2$ and Fe/W–CeO$_2$, it is known that the peak intensity near 2θ = 20° − 35° is enhanced with the load of iron and tungsten in Figure S1. Therefore, it is reasonable to assume that a new phase of FeWO$_4$ formed with the addition of Fe/W species. It was clearly found that diffraction peaks of ferric

oxide and tungsten oxide were not detected in the materials. According to the results of XRD and HRTEM, it was obtained that $Fe_2O_3$ and $FeWO_4$ species might be highly dispersed on the surface of $CeO_2$ support.

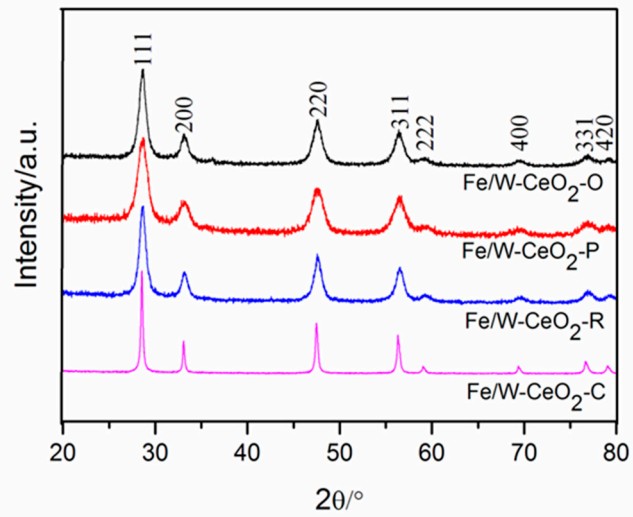

**Figure 2.** XRD patterns of Fe/W–CeO$_2$ catalysts.

**Table 1.** Physical properties of Fe/W–CeO$_2$ catalysts.

| Samples | $S_{BET}$ [a] (m$^2$ g$^{-1}$) | $V_{BJH}$ [b] (cm$^3$ g$^{-1}$) | $D_{BJH}$ [c] (nm) | Crystallite Size [d] (nm) | $S_{t\text{-plot}}$ [e] (m$^2$ g$^{-1}$) |
|---|---|---|---|---|---|
| Fe/W–CeO$_2$–O | 67.03 | 0.21 | 10.69 | 9.5 | 7.35 |
| Fe/W–CeO$_2$–P | 80.06 | 0.09 | 3.51 | 6.3 | 9.84 |
| Fe/W–CeO$_2$–R | 47.05 | 0.24 | 16.04 | 10.3 | 5.91 |
| Fe/W–CeO$_2$–C | 15.75 | 0.12 | 28.95 | 36.5 | 2.15 |

[a] BET surface area; [b] BJH desorption pore volume; [c] Average pore diameter.; [d] Calculated from the ceria (111) plane using Scherrer's formula.; [e] t-plot micropore area.

### 2.3. BET

It was seen that all samples could be divided into the type IV adsorption-desorption isotherm (Figure 3), revealing their mesoporous characteristics, which could be obtained from the packing of the nanoparticles. Fe/W–CeO$_2$–O, Fe/W–CeO$_2$–R, and Fe/W–CeO$_2$–C displayed H3 type hysteresis loop and verifed the existence of slit-shaped pores [26,27]. Especially, Fe/W–CeO$_2$–P also possessed an H4-type hysteresis loop at a relatively lower pressure, which can be explained by the interaction of cerium and iron leading to the formation of micropores. Additionally, the BET surface areas, pore volumes, and average pore diameter of the samples are summarized in Table 1. The pore distribution of Fe/W–CeO$_2$–P catalytic showed the least average pore size of 0.09 nm and the least average pore diameter of 3.51 nm, which indicated that Fe/W–CeO$_2$–P had porous structures. The porosity observed may be the interaction of iron and strontium to form some micropores for presence of intercrystalline voids between the nanocrystallites, which can be demonstrated by the values of micropore surface area. Moreover, Fe/W–CeO$_2$–P had the larger BET surface area and micropore surface area, resulting in high dispersion of the metal oxide composite with Fe/W [28]. The excellent dispersion of Fe/W–CeO$_2$–P was responsible for the smaller crystal size, consistent with HRTEM and XRD.

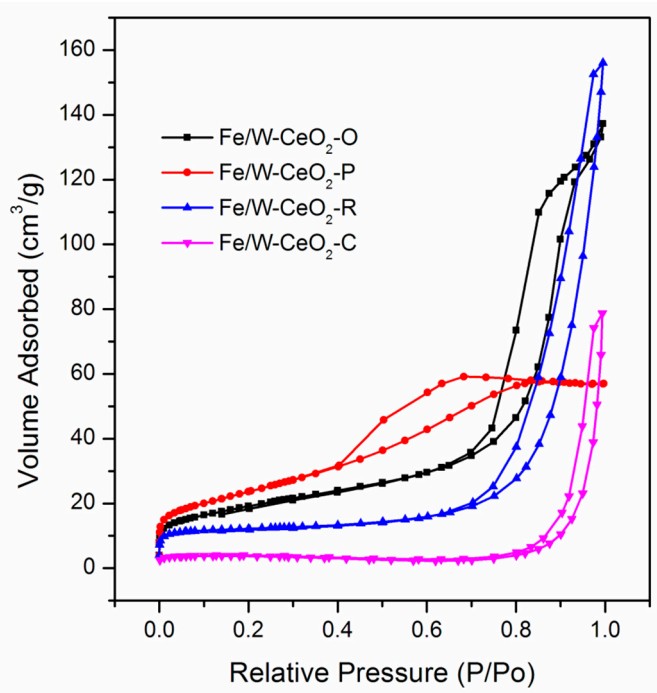

**Figure 3.** Nitrogen adsorption/desorption isotherms of Fe/W–CeO$_2$.

### 2.4. Raman

Figure 4 displays the visible Raman spectra of CeO$_2$ and Fe/W–CeO$_2$ catalysts. For the CeO$_2$ support, the distinct F$_{2g}$ symmetry mode of the CeO$_2$ phase centered at 463 cm$^{-1}$ with weak bands at 251, 600 and 1050 cm$^{-1}$ assigned to the second-order transverse acoustic (2TA), defect-induced (D), and second-order longitudinal (2LO), respectively [29–31]. The Raman spectra for the Fe/W–CeO$_2$ structure almost remained after the addition of Fe and W species in Figure 4b. In addition, the peak at 251, 463, and 590 cm$^{-1}$ in CeO$_2$ shifted to 247, 460, and 594 cm$^{-1}$ in Fe/W–CeO$_2$, respectively. The peak at 1050 cm$^{-1}$ shifted to 1170 cm$^{-1}$ and was broadened. These shifts and broadening were attributed to the Fe or W addition into the CeO$_2$ lattice, which was related to the existence of reduced states of cerium. The results further implied the combination of Fe or W ions into the surface/subsurface of ceria without resulting in a change in the original cubic structure, which was consistent with XRD and TEM results.

Furthermore, the I$_{600}$/I$_{460}$ values (the ratio of the intensities of the D and F2g bands) indicated the oxygen vacancies concentration (Figure 4c). The oxygen vacancies concentration followed the order: CeO$_2$–P > CeO$_2$–O > CeO$_2$–C > CeO$_2$–R. After the impregnation of Fe and W, all of the I$_{600}$/I$_{460}$ values increased significantly. Moreover, the amount sequence of the oxygen vacancies over Fe/W–CeO$_2$ was consistent with the similar trend of pure CeO$_2$. Consequently, Fe or W insertion into the CeO$_2$ lattice might induce the increase in defect concentration in the nanostructure ceria [32].

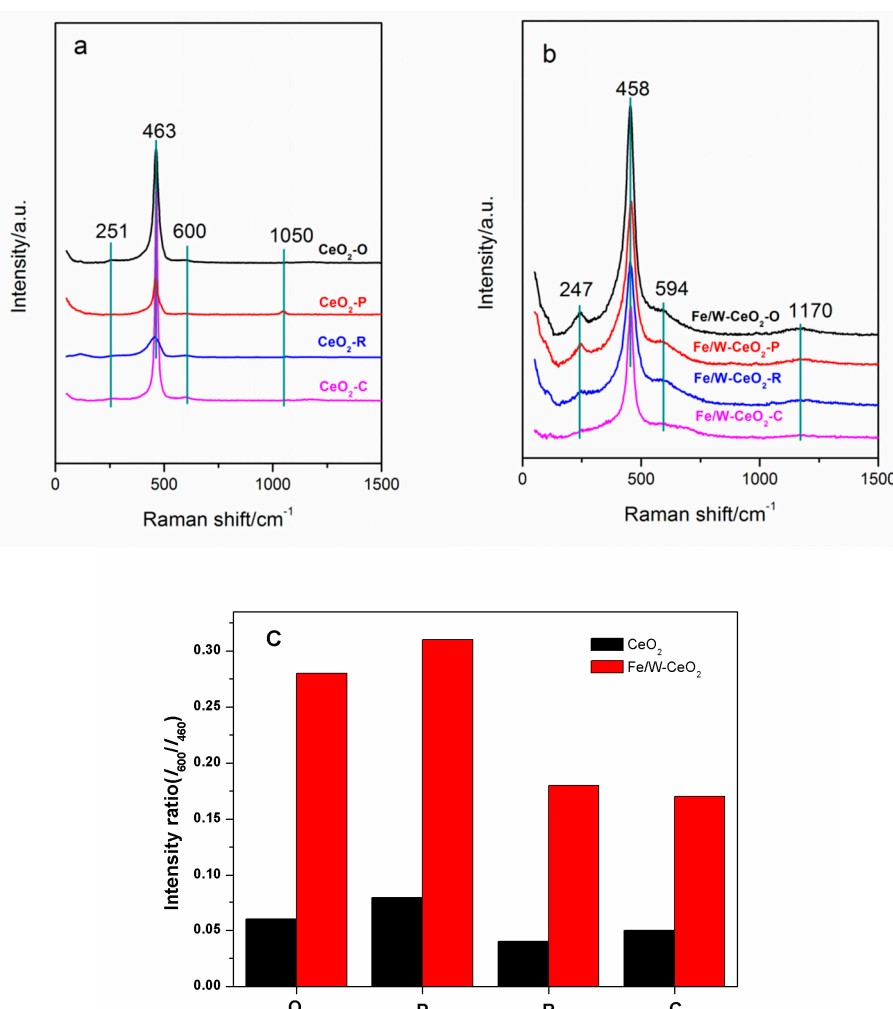

**Figure 4.** Raman spectra of (**a**) $CeO_2$, (**b**) Fe/W-$CeO_2$ ordinary, nanopolyhedra, nanorods, nanocubes, and (**c**) the corresponding peak intensity ratios $I_{600}/I_{460}$.

*2.5. XPS*

Ce3d spectra are presented in Figure 5a. The $Ce^{4+}$ has been matched with six peaks: v (882.2 eV), $v_2$ (889.5 eV), $v_3$ (898.2 eV), u (901.2 eV), $u_2$ (907.7 eV) and $u_3$ (916.7 eV). $Ce^{3+}$ has been fitted with two peaks: $v_1$ (885.1 eV) and $u_1$ (903.6 eV) [33]. The surface atomic concentrations of $Ce^{3+}$ to the total $Ce^{3+}$ and $Ce^{4+}$ on the Fe/W–$CeO_2$–O, Fe/W–$CeO_2$–P, Fe/W–$CeO_2$–R and Fe/W–$CeO_2$–C were 0.18, 0.20, 0.15, and 0.14, respectively. This indicated that the oxygen vacancies of the Fe/W–$CeO_2$–P was more than those of Fe/W–$CeO_2$–O, Fe/W–$CeO_2$–R, and Fe/W–$CeO_2$–C, which is in agreement with the result of the oxygen vacancies by Raman. Oxygen vacancies can be generated by the transformation between $Ce^{3+}$ and $Ce^{4+}$ according to the following: $4Ce^{4+} + O^{2-} \rightarrow 4Ce^{4+} + 2e^-/\square + 0.5O_2 \rightarrow 2Ce^{4+} + 2Ce^{3+} + \square + 0.5O_2$, where $\square$ represented an empty position (anion-vacant site) originating from the removal of $O^{2-}$ from the lattice [34]. The higher the $Ce^{3+}$ concentration to the total $Ce^{3+}$ and $Ce^{4+}$ present, the more oxygen vacancies formed.

Figure 5b shows the characteristic peaks of the low binding energy iron-containing catalyst assigned to Fe $2p_{3/2}$. All the catalysts showed the Fe $2p_{3/2}$ peak and the corresponding peak at 714.7–715.2 eV and 717.8–718.4 eV, which was assigned to $Fe^{2+}$ and $Fe^{3+}$, respectively [26,35]. The $Fe^{2+}$ sites were formed by the interaction between $Fe_2O_3$ and $CeO_2$ support, probably through an interfacial redox process: $xFe_2O_3 + (2 − y)CeO_{2−x} \rightarrow xFe_2O_{3−y} + (2 − y)CeO_2$ [36]. According to the calculation of $Fe^{2+}/(Fe^{2+} + Fe^{3+})$ (Table 2), the concentration of $Fe^{3+}$ decreased according to the sequence: Fe/W–$CeO_2$–P > Fe/W–$CeO_2$–O > Fe/W–$CeO_2$–C > Fe/W–$CeO_2$–R. Notably, Fe/W–$CeO_2$–P proved

obviously more $Fe^{2+}$ species than other Fe/W–$CeO_2$, indicating there was strong Fe–Ce interfacial interaction over nanopolyhedrons $CeO_2$ support [37]. Moreover, these peaks were shifted to higher binding energy compared with that of pure $Fe_2O_3$ species, which was ascribed to the strong electron interaction between iron and tungsten. W4f spectra are presented in Figure 5c. Similar peaks of binding energy of $W4f_{7/2}$ (34.6–34.9 eV) and $W4f_{5/2}$ (36.6–36.9 eV) were observed over all the Fe/W–$CeO_2$ samples. These binding energies on Fe/W–$CeO_2$ catalysts were indeed lower than the corresponding values on original $W^{6+}$ sample ($W4f_{7/2}$ at 35.8 eV and $W4f_{5/2}$ at 38.0 eV) [38,39]. This phenomenon should be related to the formation of $FeWO_4$ species (Figure 1) with electronic defects, which resulted in the increment of electron cloud of $W^{6+}$ species. Feng et al. [40] also observed that the as-prepared monolayer-dispersed $WO_x$ species on $\alpha$-$Fe_2O_3$ surface had mixed oxidation states of $W^{6+}$ and $W^{5+}$.

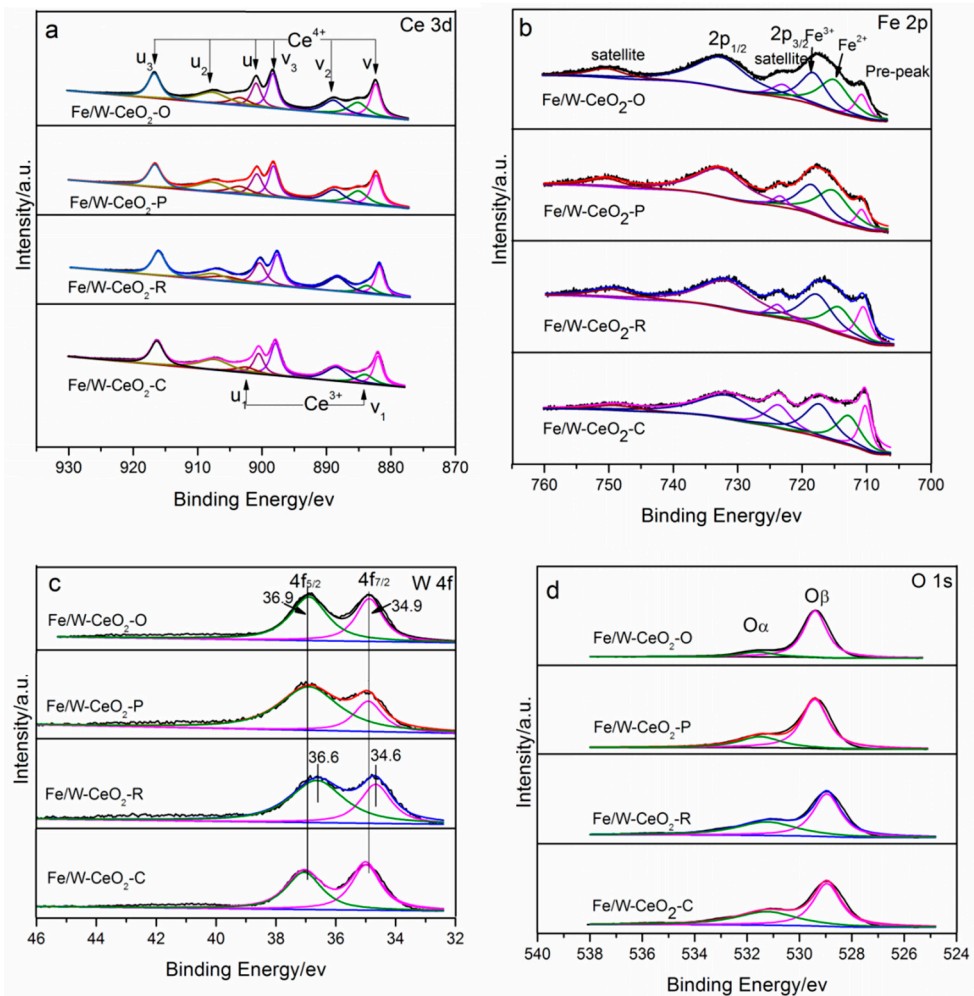

**Figure 5.** XPS spectra of (**a**) Fe 2p, (**b**) Ce 3d, (**c**) W 4f, and (**d**) O 1s for Fe/W–$CeO_2$ catalysts.

**Table 2.** Surface atomic concentration.

| Samples | Fe/W | $Ce^{3+}/(Ce^{3+} + Ce^{4+})$ | $O_\alpha/(O_\alpha + O_\beta)$ | $Fe^{2+}/(Fe^{2+} + Fe^{3+})$ |
|---|---|---|---|---|
| Fe/W–$CeO_2$–O | 0.073 | 0.18 | 0.13 | 0.54 |
| Fe/W–$CeO_2$–P | 0.092 | 0.20 | 0.26 | 0.56 |
| Fe/W–$CeO_2$–R | 0.083 | 0.15 | 0.41 | 0.43 |
| Fe/W–$CeO_2$–C | 0.063 | 0.14 | 0.27 | 0.42 |

## 2.6. UV–VIS Diffuse Reflectance Spectra

Figure 6 reveals the UV–VIS absorption spectra of Fe/W–CeO$_2$ catalysts. Subbands around 215 and 276 nm arise from isolated Fe$^{3+}$ sites in tetrahedral and higher coordination (five or six oxygen ligands), respectively. A broad band between 300 and 400nm in all catalyst can be deconvoluted into subbands at 316 nm, which was assigned to small oligomeric Fe$_x$O$_y$ clusters. In addition, an additional shoulder peak at around 365 nm was attributed to WO$_3$ [41], which further proofed of the XPS results about the W$^{6+}$ existence. Meanwhile, subbands > 400 nm (centered at around 521 nm) was assigned to large Fe$_2$O$_3$ particles [42], which was consistent with HRTEM results. This peak can be related to the degree of dispersion of metal. The stronger intensity absorption band of Fe$_2$O$_3$ was detected, suggesting that iron was in highly dispersive state, no large iron oxide particles aggregated in the sample [43,44]. It was noted that the Fe/W–CeO$_2$–P catalyst showed the weakest intensity peak of Fe$_2$O$_3$, suggesting the highest dispersion of iron in Fe/W–CeO$_2$–P catalyst.

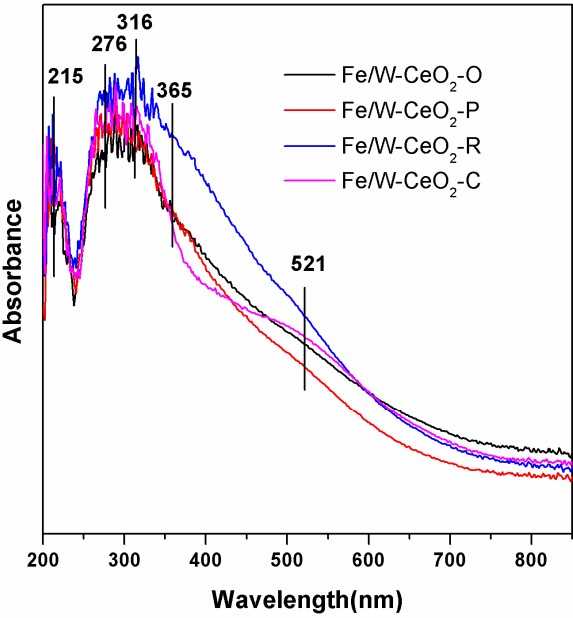

**Figure 6.** UV–VIS of Fe/W–CeO$_2$ catalysts.

## 2.7. Py-IR Spectra

Figure 7 shows the Py-IR spectra of three catalysts. Among these, the observed IR bands at 1441, 1574 and 1597 cm$^{-1}$ correspond to different modes of vibration of pyridine coordinated to Lewis acid sites. The band at 1490 cm$^{-1}$ was characteristic of the Lewis–Brønsted acid complex [45]. According to Emeis and Zhang et al. [46,47], the concentration of acid sites on four Fe/W–CeO$_2$ were calculated. In addition, the calculated results and the concentration of total acid sites of four catalysts followed the sequence of Fe/W–CeO$_2$–P (223 mmol/g) > Fe/W–CeO$_2$–O (207 mmol/g) > Fe/W–CeO$_2$–C (104 mmol/g) > Fe/W–CeO$_2$–R (31 mmol/g).

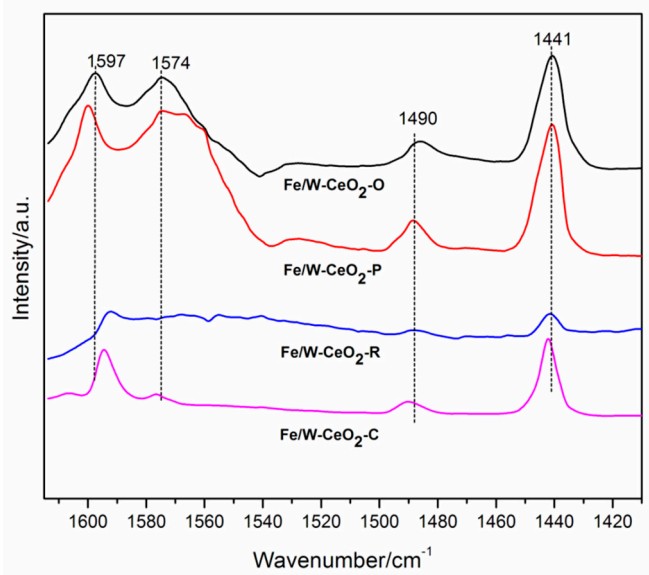

**Figure 7.** Py-IR spectra of four Fe/W–CeO$_2$ catalysts.

## 2.8. Activity Measurement

As is shown in Figure 8, the catalytic activity of NH$_3$-SCR on different morphological catalysts was investigated. It was obvious that the NO$_x$ conversion over all Fe/W–CeO$_2$ increased first and decreased with the increase of temperature. Especially, the Fe/W–CeO$_2$–R line sharp rose from 20% up to 82% (the vertex) of NO$_x$ conversion, which sudden dropped at 350 °C. The Fe/W–CeO$_2$–P displayed the highest NO$_x$ conversion and the widest range of operating-temperature window between 225 and 325 °C. Among the four catalysts, the NO conversion efficiency decreased in the order: Fe/W–CeO$_2$–O > Fe/W–CeO$_2$–R > Fe/W–CeO$_2$–C. As shown in Figure 8b, all Fe/W–CeO$_2$ catalysts showed 100% N$_2$ selectivity below 225 °C. By increasing the reaction temperature, the N$_2$ selectivity decreased slightly, but it was still higher than 95%.

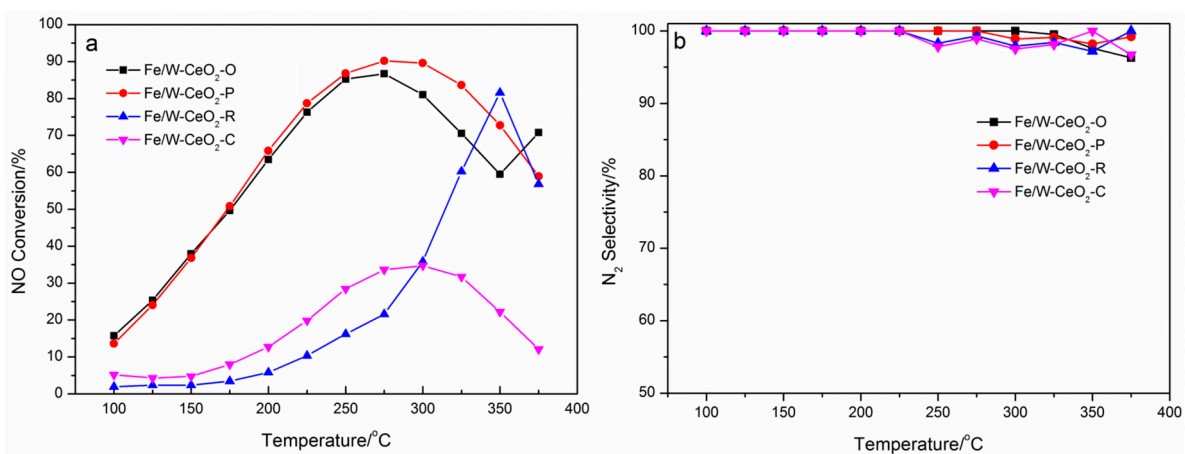

**Figure 8.** Different morphology Fe/W–CeO$_2$ catalysts of (**a**) NO conversions and (**b**) N$_2$ selectivity. Reaction conditions: [NO] = [NH$_3$] = 500 ppm, [O$_2$] = 3%, GHSV = 50,000 h$^{-1}$ and Ar as balance gas.

The NO reaction rates for Fe/W–CeO$_2$–O, Fe/W–CeO$_2$–P, Fe/W–CeO$_2$–R, and Fe/W–CeO$_2$–C are $1.88 \times 10^{-4}$ mol$_{NO}$ g$_{cat}$$^{-1}$ s$^{-1}$, $2.40 \times 10^{-4}$ mol$_{NO}$ g$_{cat}$$^{-1}$ s$^{-1}$, $2.34 \times 10^{-5}$ mol$_{NO}$ g$_{cat}$$^{-1}$ s$^{-1}$, and $3.97 \times 10^{-5}$ mol$_{NO}$ g$_{cat}$$^{-1}$ s$^{-1}$ at 150 °C, respectively. This value were in the order: Fe/W–CeO$_2$–P > Fe/W–CeO$_2$–O > Fe/W–CeO$_2$–C > Fe/W–CeO$_2$–R, representing that CeO$_2$ (111) was superior to CeO$_2$ (110) and (100) as a carrier for FeWO$_x$.

An Arrhenius plot in view of reaction rate data between 100 and 150 °C is displayed in Figure 9. The apparent activation energy of Fe/W–CeO$_2$–P was 36 kJ mol$^{-1}$, which is lower than that of Fe/W–CeO$_2$–O, Fe/W–CeO$_2$–R and Fe/W–CeO$_2$–C catalysts (41, 55 and 48 kJ mol$^{-1}$). Therefore, the catalytic performance of Fe/W–CeO$_2$–P catalyst was better than that of the Fe/W–CeO$_2$–O, Fe/W–CeO$_2$–R, and Fe/W–CeO$_2$–C catalysts. According to the references [48,49], it can draw a conclusion that the lower the apparent activation energy, the easier the reaction, so the Fe/W–CeO$_2$–P catalyst requires less energy in the reaction, indicating the excellent catalytic performance.

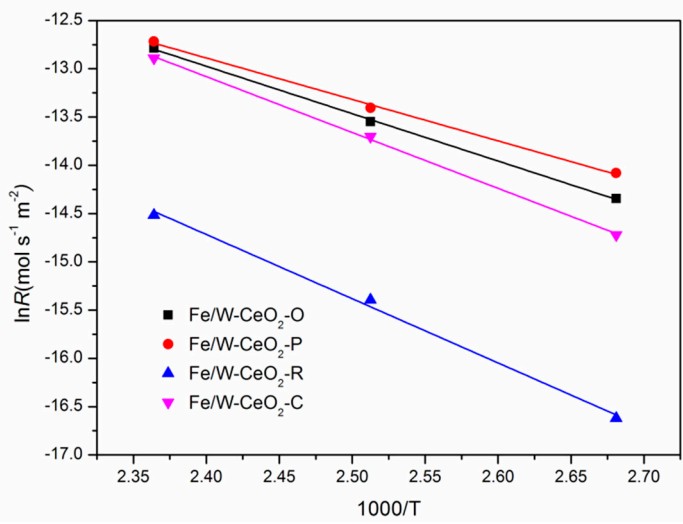

**Figure 9.** Arrhenius plots of the inherent reaction rate constants of Fe/W–CeO$_2$ catalysts.

## 2.9. The Effect of SO$_2$ and H$_2$O

The effect of extra gas streams of 200 ppm SO$_2$ and/or 5% H$_2$O on de-NO$_x$ measurement with a constant temperature of 300 °C are shown in Figure 10. When the additional gas stream was turned on and off, the de-NO$_x$ measurement remained constant for a few minutes. The NO conversion reduced from 90% to 86% after adding SO$_2$. After the addition of H$_2$O, the NO conversion reduced to 82%, and when SO$_2$ and H$_2$O were added at the same time, the NO conversion was about 75%. After removing the additional gas, in all three cases, the NO conversion returned to a value very close to the initial value. The above results indicated that Fe/W–CeO$_2$ showed excellent resistance of sulfur dioxide and water, which may be due to the addition of iron and tungsten mentioned in the literature. The addition of iron oxide significantly reduced the rate of sulfate formation, thereby inhibiting the progress of the reaction [50]. For tungsten oxide, it can reduce the stability of cerium sulfate, which occupied the active site of catalyst [51].

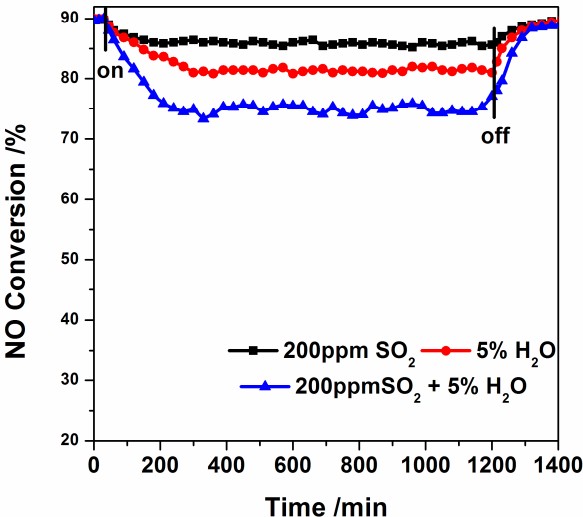

**Figure 10.** The effect of $SO_2$ and $H_2O$ on NO conversion over Fe/W–CeO$_2$–P at 300 °C. Reaction conditions: [NO] = [NH$_3$] = 500 ppm, [O$_2$] = 3%, 200 ppm SO$_2$ (when used), 5 vol.% H$_2$O (when used), GHSV = 50,000 h$^{-1}$, and Ar as balance gas.

## 3. Discussion

The activities of four Fe/W–CeO$_2$ can be ranked: Fe/W–CeO$_2$–P > Fe/W–CeO$_2$–O > Fe/W–CeO$_2$–C > Fe/W–CeO$_2$–R at temperatures of 100–300 °C. On the one hand, the catalytic behavior of Fe/W–CeO$_2$–P and Fe/W–CeO$_2$–O is significantly different from that of Fe/W–CeO$_2$–R and Fe/W–CeO$_2$–C on the other hand. The former two are more active and their conversion curves cover the temperature range of 100 to 250 °C, although the Fe/W–CeO$_2$–P based catalyst has higher activity in the range of 250–300 °C. Both Fe/W–CeO$_2$–P and Fe/W–CeO$_2$–O have too many similarities in terms of morphology and exposed crystal faces. Therefore, in the subsequent oxygen vacancies, the BET specific surface area, the dispersion of iron oxide, and the amount of acid all have close performance, which is also reflected in the catalyst activity in the low temperature section. Additionally, the crystal faces may not be the main influencing factors of both. However, the difference in the preparation methods between Fe/W–CeO$_2$–P and Fe/W–CeO$_2$–O may be the main influencing factors [52–54], and the incomplete agreement among the exposed crystal faces leads to a significant difference in the catalytic activity in the high temperature range.

According to the above-mentioned characterization results, Fe/W–CeO$_2$–P catalyst possessed higher oxygen vacancies (XPS and Raman). In addition, Fe/W–CeO$_2$–P was exposed to more stable crystal plane with CeO$_2$ (111) plane (HRTEM) where low cation such as Ce$^{3+}$ and Fe$^{2+}$ was more likely to exist because they are not easily oxidized. For details, the spontaneous reduction of Ce$^{4+}$ to Ce$^{3+}$ was known to improve the formation of oxygen vacancies for maintaining electrostatic balance [55]. It was generally believed that oxygen migration in ceria-based materials was carried out through vacancy hopping mechanism [56]. Thus, Fe/W–CeO$_2$–P catalyst with the oxygen vacancies might was beneficial to oxygen migration and has more active oxygen than other catalysts.

The higher dispersion of iron in Fe/W–CeO$_2$–P catalyst are received by the results of XRD and UV–VIS. The a new active phase of Fe and Ce can enhanced the NO$_x$ reduction performance [57]. By XPS, Fe/W–CeO$_2$–P present the higher Fe$^{2+}$ concentration or a litter amount of Fe$_2$O$_3$, which may due to stable CeO$_2$ (111) plane. X-ray diffraction and UV–VIS spectroscopy results support the idea that a large amount of Fe$_2$O$_3$ in three other Fe/W–CeO$_2$ catalysts may cover the active side of CeO$_2$. This may explain the Fe/W–CeO$_2$–P good performs as one factor.

The BET results indicated that Fe/W–CeO$_2$–P had the relatively highest BET surface area and micropore surface area (Table 1), which could improve the strong synergism between iron and cerium (XPS) and the dispersion of Fe$_2$O$_3$ (UV–VIS). The acid sites in Fe$_2$O$_3$ could adsorb NH$_3$ and NO by

Langmuir-Hinshelwood mechanism. Then adsorbed species could be converted into corresponding intermediates, such as coordination $NH_3$ and $NH^{4+}$ iron [58]. Therefore, the Fe/W–$CeO_2$–P with most amount of acid sites showed the best $NH_3$-SCR reaction performance.

## 4. Materials and Methods

### 4.1. Catalysts Preparation

The all chemicals were purchased from Aladdin Industrial Corporation (shanghai, China). The $CeO_2$ were prepared by a hydrothermal method, using $Ce(NO_3)_3 \cdot 6H_2O$ as the source of cerium and NaOH as the precipitant. The morphologies of $CeO_2$ nano-particles were governed by changing the concentration of NaOH, and the temperature of synthesis. The details in (1) $Ce(NO_3)_3 \cdot 6H_2O$ and NaOH were each dissolved in homemade deionized water. Then the above two solutions were mixed in a large beaker and kept stirring for 30 min to form the emulsion. Subsequently, the emulsion was poured into a Teflon bottle and hydrothermally treated at a temperature of 100 °C for 24 h, denoted as B; (2) The second catalyst only changes the concentration of NaOH to 6mol/L. The other conditions are the same as those of the B catalyst, denoted as C; (3) The third catalyst only adjusts the hydrothermal temperature to 180 °C. The other conditions are the same as the C catalyst preparation conditions, denoted as D. Then after the hydrothermal treatment, all fresh precipitates were separately separated by centrifugation, washed two or three times with homemade deionized water and pure ethanol, and dried overnight in air at 60 °C; (4) the $Ce(NO_3)_3 \cdot 6H_2O$ was directly calcined in static air by muffle furnace at 500 °C for 3 h at a ramping rate of 2 °C $min^{-1}$ as the A catalyst. $Fe(NO_3)_3 \cdot 9H_2O$ was dissolved in the adequate oxalic acid ($H_2C_2O_4 \cdot 2H_2O$) solution, then ammonium metatungstate (($NH_4)_6H_2W_{12}O_{40} \cdot xH_2O$) was added into the solution for W in portion. The loading of active phase F/W of the main four catalysts was 9.5 to 1.0 of molar ratio. These $CeO_2$ powders were impregnated with incipient wetness by the solution and dried at 70 °C overnight, which were calcined in static air by muffle furnace at 550 °C for 3 h with a ramping rate of 2 °C $min^{-1}$ subsequently [59,60].

### 4.2. Characterizations of Catalysts

X-ray diffraction (XRD) measurements were performed on a Bruker (Billerica, MA, USA) AXS-D8. The advanced powder diffractometer is equipped with monochrome detector and CuKa radiation source. The diffraction patterns were recorded in the 2θ range of 20°–80° with a step size of 0.02°. The morphologies of the as-prepared samples were characterized by a field emission transmission electron microscope (TEM) of Tecnai G2 F20 (FEI, Hillsboro, OR, USA), dispersed into analytical purity alcohol with an ultrasound system before observation. Nitrogen adsorption/desorption measurements of the samples were accomplished on a surface analyzer (Micrometrics, ASAP 2020, Norcross, GA, USA) at 77K. The samples specific surface area was calculated using the Brunauer–Emmett–Teller (BET) method. The pore size distributions were determined from desorption branches by the Barrett–Joyner–Halenda (BJH) method. The chemical states of the atoms on the surface of the catalyst were investigated using a VG ESCALAB 210 electron spectrometer (Mg Kα radiation; hν = 1253.6eV, Thermo Fisher Scientific, Waltham, MA, USA). The XPS data were calibrated using the C1s (284.6 eV) binding energy as the standard. UV–VIS diffuse reflectance spectra were collected on a Shimadzu (Kyoto, Japan) UV 2550 spectrophotometer. The FTIR of pyridine (Py) was measured on Thermo Nicolet 6700 FTIR Spectrometer (Thermo Fisher Scientific, Waltham, MA, USA) with a resolution of 0.5 $cm^{-1}$.

### 4.3. Activity Measurement

The catalytic performance by a fixed-bed quartz tube reactor was evaluated. The gas composition of the 0.5 g Fe/W–$CeO_2$ sample was as follows: 500 ppm NO, 500 ppm $NH_3$, 3 vol.% $O_2$, Ar balance, and a total flow rate of 300 mL/min. Hence the corresponding gas hourly space velocity (GHSV) was 50,000 $h^{-1}$. The catalysts were pressed, comminuted and sieved to 20–40 mesh prior to each activity test,

then filled into a quartz pipe reactor with an inner diameter of 8 mm. At each temperature step of 25 °C between 100 to 375 °C, the concentrations of NO, $NH_3$ and $N_2O$ were measured. The concentrations of NO in the inlet and outlet gases were measured using a 4000VM NOx analyzer. The concentrations of $N_2O$ and $NH_3$ were measured by a G200 analyzer and an IQ 350 ammonia analyzer.

NO conversion was calculated according to the following equations:

$$NO\ Conversion\ (\%) = \frac{[NO]_{in} - [NO]_{out}}{[NO]_{in}} \times 100\% \tag{1}$$

$$N_2\ Selectivity\ (\%) = (1 - \frac{2[N_2O]_{out}}{[NO]_{in} - [NO]_{out} + [NH_3]_{in} - [NH_3]_{out}}) \times 100\% \tag{2}$$

In order to comprehend the intrinsic influence of the morphology of $CeO_2$ nanostructures on the Fe/W-$CeO_2$, the catalytic performance data were analyzed by the macroscopic kinetics. The $NH_3$-SCR on the Fe/W-$CeO_2$ was generally recognized to be a first-order reaction related to NO [61,62]. Kinetic experiments were implemented in a fixed-bed reactor. The NOx conversion was modified to below 40% so as to eliminate the thermal effect and diffusion effect before calculating the reaction rate which was normalized over the surface area and shown below:

$$k = \frac{FX}{W_c S_c} \tag{3}$$

where $k$ is the reaction constant (mol s$^{-1}$ m$^{-2}$), $F$ is the molar NO feed rate (mol s$^{-1}$), $X$ is the conversion of NO, $W_c$ is the catalyst weight (g), (%), $S_c$ is the catalyst BET surface area, and T is the temperature ($K$). The apparent activation energies ($E_a$) were calculated by the Arrhenius law (ln ($k$) = f (1000/T)).

## 5. Conclusions

Four Fe/W–$CeO_2$ with different ceria morphologies as supports were prefabricated by a hydrothermal method. The HRTEM images revealed that Fe/W–$CeO_2$–P, Fe/W–$CeO_2$–R, and Fe/W–$CeO_2$–C mainly exposed $CeO_2$ (111), (110), and (100) crystal planes, respectively. Characterization results showed that the morphology of $CeO_2$ support originated from selectively exposing different crystal surfaces. The Fe/W–$CeO_2$–P catalyst with more oxygen vacancies, more acid sites, the higher BET surface area and the best dispersion of $Fe_2O_3$, which exhibited better activity of $NH_3$-SCR. The $CeO_2$ (111) was the most inert crystal plane compared to the well-known reactive (110) and $CeO_2$ (100) planes. Therefore, the interface behavior and catalytic characteristics of Fe/W–$CeO_2$ catalysts for NO reduction was mainly on account of the morphology and crystal-plane effect and the synergistic effect between active species and supports.

**Supplementary Materials:** The following are available online at http://www.mdpi.com/2073-4344/9/3/288/s1, **Figure S1.** XRD patterns of Fe/W-$CeO_2$ and $CeO_2$ catalysts.

**Author Contributions:** Z.W., X.L., and F.C. conceived and designed the experiments; Z.W. and D.C. revised the paper; F.L. and D.W. analyzed the data; F.L. performed the experiments, and wrote the paper.

**Funding:** The work described above was supported by the National Natural Science Foundation of China (no: 21607162, 2157061214, 21676287), and the Natural Science Foundation of Shandong Province (no. ZR2017ZC0633).

**Conflicts of Interest:** The authors declare no conflict of interest.

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
