# Peer review of "Morphology and Crystal-Plane Effects of Fe/W-CeO2 for Selective Catalytic Reduction of NO with NH3"

_catalysts, doi:10.3390/catal9030288_

Round 1

Reviewer 1 Report

Reviewer’s Report

Morphology and Crystal-Plane Effects of Fe/W-CeO2 for selective catalytic reduction of NO with NH3

Manuscript ID: catalysts-448638

The manuscript by Liu et al. investigated the Fe/W-CeO2 catalysts for NH3-SCR reaction. They reported that the morphology of CeO2 support originated from selectively exposing different crystal surfaces had a significant effect on oxygen vacancies, acid sites, and the dispersion of Fe2O3, thereby on the NH3-SCR performance. The presented results could be publishable after major revisions.

1)      Generally, the redox properties of the catalysts are important for the NH3-SCR reaction. Therefore, it is necessary to characterize the catalysts by H2-TPR analysis and the following references could be useful for the discussion.

(Journal of Catalysis 338 (2016) 56–67; Ind. Eng. Chem. Res. 2017, 56, 1772−1781; Molecular Catalysis 451 (2018) 20–32)

2)      Although the Fe/W-CeO2-O didn’t show crystal planes in HRTEM image, it showed clear crystal planes in powder XRD. Authors should explain the reason for this observation.

3)      The oxygen vacancies trend obtained from Raman analysis is different from XPS spectra. However, visible Raman spectroscopy gives information from both the bulk and the surface of the sample because of weak absorption. Therefore, how can you justify the difference in the oxygen vacancies trend from Raman and XPS analyses?

4)      Authors should provide the stability and SO2/H2O resistance of the best catalyst.

5)      It would be better to provide the Fe dispersion on the W-CeO2 support using EDS color mappings.

6)      There are some recent good papers on the CeO2-based catalysts and NOx removal catalysis that are not cited (Montini et al. Chem. Rev. 116, 5987–6041; Devaiah et al. Catal Rev 60 (2018) 177–277; Boningari et al. Journal of Catalysis 365 (2018) 320-333; Damma et al. Industrial & Engineering Chemistry Research 57 (2018) 16615-16621). 

Author Response

The manuscript by Liu et al. investigated the Fe/W-CeO2 catalysts for NH3-SCR reaction. They reported that the morphology of CeO2 support originated from selectively exposing different crystal surfaces had a significant effect on oxygen vacancies, acid sites, and the dispersion of Fe2O3, thereby on the NH3-SCR performance. The presented results could be publishable after major revisions.

1) Generally, the redox properties of the catalysts are important for the NH3-SCR reaction. Therefore, it is necessary to characterize the catalysts by H2-TPR analysis and the following references could be useful for the discussion.

(Journal of Catalysis 338 (2016) 56-67; Ind. Eng. Chem. Res. 2017, 56, 1772-1781; Molecular Catalysis 451 (2018) 20-32)

Response:

     Thanks for your comment. The hydrothermal synthesis of the catalyst used in this manuscript is directly used for catalytic reaction. The active component of catalysis is metal oxide, which does not need hydrogen reduction. In addition, XPS test has indicated the valence distribution of each component in the catalyst, so we do not think it necessary to analysis by H2-TPR.

2) Although the Fe/W-CeO2-O didn’t show crystal planes in HRTEM image, it showed clear crystal planes in powder XRD. Authors should explain the reason for this observation.

Response:

    Thanks for the reviewer’s comment. We tested all morphologies of the catalysts, including the Fe/W-CeO2-O, using HRTEM and XRD. Fe/W-CeO2-O also exposes crystalline planes and is similar to polyhedrons, but it also exposes other crystalline planes without definite crystalline planes. Thank you again for your comment. We has revised it, the corresponding modifications were found in line 68 to 70 on Page 2 in the revised manuscript.

3) The oxygen vacancies trend obtained from Raman analysis is different from XPS spectra. However, visible Raman spectroscopy gives information from both the bulk and the surface of the sample because of weak absorption. Therefore, how can you justify the difference in the oxygen vacancies trend from Raman and XPS analyses?

Response:

    Thanks for your comment. Oxygen vacancies played an important role in NH3-SCR process. As you can see, Raman spectroscopy gave information from both bulk and surface of the sample because of weak absorption. XPS gave information from the surface of the sample. The NH3-SCR reaction takes place on the surface of the catalyst, so we believe that the reaction is more closely related to the oxygen vacancies on the surface. Anyway, the trend of catalyst activity is the same as that of surface oxygen vacancy, which we have more reason to believe. The corresponding revision can be found in line 249 to 251 on Page 11 in the revised manuscript.

4) Authors should provide the stability and SO2/H2O resistance of the best catalyst.

Response:

    Thanks for your good comment. As is well-known, there is still residual H2O and SO2 in the flue gas even with the desulfurization before de-NOx measurement. Therefore, it is essential to investigate the eect of H2O and SO2 on the NH3-SCR activity of catalysts. We carefully and urgently made relevant experiments. The corresponding revision can be found in line 223 to 238 on Page 10 with Figure 10 in the revised manuscript.

5) It would be better to provide the Fe dispersion on the W-CeO2 support using EDS color mappings.

Response:

    Thanks for your professional comment. To test the distribution of elements in catalysts, EDS color mapping should be done. This is a good suggestion. We are sorry not to do it, for  laboratory conditions are not allowed to do it. In the following, we will create conditions to explore EDS color mapping.

6)      There are some recent good papers on the CeO2-based catalysts and NOx removal catalysis that are not cited (Montini et al. Chem. Rev. 116, 5987–6041; Devaiah et al. Catal Rev 60 (2018) 177–277; Boningari et al. Journal of Catalysis 365 (2018) 320-333; Damma et al. Industrial & Engineering Chemistry Research 57 (2018) 16615-16621).

Response:

    Thanks for your comment. This manuscript didn’t have enough references. It needs better references to enrich it.  The corresponding revision can be found in line 36 on page 1, line 45, 49 on page 2 and line 278 on page 12 with references [14], [17], [18] and [54].

Reviewer 2 Report

General comments.

Low temperature selective catalytic reduction (SCR) of NOx with NH3 is a topic of high interest and of interest to the readers of Catalysts. The authors present a comparative study of different Fe/W-CeO2 catalysts with different morphology and fase exposure of the CeO2 support. Although the characterization of the materials has been performed in a thorough and systematic way, there are important contradictions and inconsistencies along the discussion, especially of the activity measurements, that should be clarified before the paper can be accepted for publication.

More details are given below.

Specific comments.

1.      English usage at some points should be revised

2.      Introduction: Additional references should be included. 

a.       Ref. 14 could be completed with Laursen et al.,  Angew. Chem. Int. Ed. 2012, 51(17), 4190-4193.

b.      Paragraph enclosed within lines 41-49 should be completed with references regarding the beneficial effect of Ce based catalysts, their active species, oxygen vacancies and metal-support interactions.

3.      The ordinary amorphous CeO2 should be better defined. It cannot be amorphous as it presents the typical XRD patter of the fluorite cubic structure. The differences between this material (CeO2-O) and the nanopolyhedrons (CeO2-P) should be clearly exposed at the beginning of the manuscript.

4.      TEM and HRTEM: in the case of the nanorods and the cubes the Fe2O3 and FeWO4 particles can be distinguished from the CeO2 support because of the different morphology. However, in the case of the ordinary CeO2 and the nanopolyhedra the authors make the distinction based on the interplanar distance, which in some cases is very similar (0.29 nm for  (111) crystal plane of FeWO4 and 0.31 nm for the (111) crystal plane of CeO2). In these two cases it would be convenient to determine the chemical composition, for instance by elastic collision (STEM) or inelastic collision based techniques (EDX, EELS).

5.      BET: as the authors mention in lines 103-104, the porosity observed is probably due to the presence of intercrystalline voids between the nanocrystallites. If some microporosity is formed as a consequence of the interaction of iron and cerium, it could be determined by means of the t-plot procedure. The external surface area could then be calculated as the difference between the BET surface area and the micropore surface area. Perhaps this would justify the differences observed for the CeO2-O and –P supports, very similar according to the TEM and XRD results.

6.      XPS: Table 2, enclosing the surface atomic concentrations, should be completed with experimentally determined bulk compositions (Fe/Ce, Fe/W), for instance by ICP.

7.      UV-Visible diffuse reflectance spectra: according to this section, Fe/W-CeO2-P presents the lowest iron dispersion as compare to the other catalysts. This is in total disagreement with the rest of the discussion and results, and is unexpected due to the small crystal size and high BET surface area of the CeO2-P support.

8.      IR-Py: Although the IR band at 1490 cm-1 is assigned to Lewis-Bronsted acid complex, the samples described here do not present any Bronsted acidity (no band is observed at 1540 cm-1), so the 1490 cm-1 band is due only to Lewis acidity.

9.      Activity measurements: There is a clear difference in the catalytic behavior of Fe/W-CeO2-P and –O, on the one hand, and Fe/W-CeO2-R and –C on the other. The former two are much more active and their conversion curves overlay in a temperature range of 100 to 250ºC. This clearly indicates a direct correlation between the crystal size of the CeO2 support and the NO conversion, and other issues such as oxygen vacancies, or Fe-Ce and Fe-W interactions do not seem to play a determining role at low temperatures. In my opinion the differences in activity between the CeO2-O and the CeO2-P based catalysts are small and do not justify the activity ranking given by the authors in section 3. This Discussion section should be completely rewritten, and clear arguments should be presented correlating the different properties of the four catalysts with their true catalytic behavior. It is important to isolate as much as possible the effects of the morphology, of the crystal size and of the crystal faces exposed by the four CeO2 supports compared on the iron dispersion, on the interaction of iron with the support, on the Fe-W interaction, on the amount of oxygen vacancies, on the acidity. And all those properties have to be correlated in a coherent way, between them and also with the catalytic properties. The authors present a large amount of information regarding the properties of the catalyst. This information has to be analyzed in a global way, not individually, one technique after the other, as it is in the present manuscript. Once the properties of the catalysts are clearly exposed, it will be easier to find correlations with the catalytic behavior.

10.  Finally, the rates given in the text (lines 207-209) are given in mol NO/g·s and the Arrhenius plot is based on the rates determined per m2 of catalyst. This is confusing and leads to false conclusions: according to this plot, catalyst Fe/W-CeO2-C should present a behavior in the 100-150ºC temperature range closer to the –P and –O catalysts than to the nano-rods based catalyst, which is not true. The plot should be given based on rates determined per gram of catalyst. If helpful for discussion, it can also be given per m2, but only as a second option.

11.  Materials and methods: This sections should be placed before the Results and Discussion section. The description of the catalyst preparation has to be improved. Please name the preparations 1, 2, 3 and 4 according to the nomenclature given in the rest of the manuscript (CeO2-O, -P, -R and –C). Is the impregnation performed by incipient wetness? Please specify. Which is the diameter of the reactor? According to section 4.3, the kinetic experiments were limited to NOx conversions below 17%. However, the conversion for the CeO2-O and –P based catalysts in the 100-150ºC range reaches values close to 40%. Are these kinetic experiments different from the ones presented in figure 8? If so, the activity and selectivity plots should also be included. Moreover, although it is important to limit conversion for accurate kinetic measurements, low conversion levels do not ensure absence of mass transfer (external and internal diffusion) control. This will be related to experimental factors such as reactor diameter, length of the catalytic bed, total flow and catalyst particle size.

Author Response

Specific comments.

1)English usage at some points should be revised.

Response:

    Thanks for the reviewer’s comment. The manuscript has been carefully revised in order to improve English writing and reduce grammatical errors in English usage. Moreover, we have looked for experts who are native speakers of English to polish this manuscript. The corresponding revision can be found in line 20 to 22 on page 1, line 26 to 31 on page 1, line 38 to 40 on page 2, line 126 to 128 on page 5, line 181 to 182 on page 8 and line 275 to 278 on page 11.

2)Introduction: Additional references should be included.

a.       Ref. 14 could be completed with Laursen et al.,  Angew. Chem. Int. Ed. 2012, 51(17), 4190-4193.

b.      Paragraph enclosed within lines 41-49 should be completed with references regarding the beneficial effect of Ce based catalysts, their active species, oxygen vacancies and metal-support interactions.

Response:

Thanks for the reviewer’s comment. The references you mentioned have been revised in the manuscript.

a. Ref. 14 have completed with Laursen et al.,  Angew. Chem. Int. Ed. 2012, 51(17), 4190-4193 called Ref. 16 in line 37 on page 1

b. Paragraph enclosed within lines 41-49 have completed with references regarding the beneficial effect of Ce based catalysts, their active species, oxygen vacancies and metal-support interactions called Ref. 18 in line 49 on page 2.

3)The ordinary amorphous CeO2 should be better defined. It cannot be amorphous as it presents the typical XRD patter of the fluorite cubic structure. The differences between this material (CeO2-O) and the nanopolyhedrons (CeO2-P) should be clearly exposed at the beginning of the manuscript.

Response:

    Thanks for the reviewer’s good comments. We have tested morphologies of the catalysts, including the Fe/W-CeO2-O, using HRTEM and XRD. Fe/W-CeO2-O also exposed crystalline planes and was similar to polyhedrons, but it also exposed other crystalline planes without definite crystalline planes. Thank you again for your comment. The corresponding modifications were as follows: line 68 to 70 on Page 2 in the revised manuscript.

4)TEM and HRTEM: in the case of the nanorods and the cubes the Fe2O3 and FeWO4 particles can be distinguished from the CeO2 support because of the different morphology. However, in the case of the ordinary CeO2 and the nanopolyhedra the authors make the distinction based on the interplanar distance, which in some cases is very similar (0.29 nm for  (111) crystal plane of FeWO4 and 0.31 nm for the (111) crystal plane of CeO2). In these two cases it would be convenient to determine the chemical composition, for instance by elastic collision (STEM) or inelastic collision based techniques (EDX, EELS).

Response:

     Thanks for your insightful comment. The ordinary CeO2 also exposed crystalline planes and was similar to polyhedrons, but it also exposed other crystalline planes without definite crystalline planes. It could make the distinction based on the interplanar distance, which can distinguish the ordinary CeO2 and the nanopolyhedra. Of course, the method provided about elastic collision (STEM) or inelastic collision based techniques (EDX, EELS) can better distinguish the two. We can achieve the same goal with the previous revision. The corresponding modifications were as follows: line 68 to 70 on Page 2 in the revised manuscript.

5)BET: as the authors mention in lines 103-104, the porosity observed is probably due to the presence of intercrystalline voids between the nanocrystallites. If some microporosity is formed as a consequence of the interaction of iron and cerium, it could be determined by means of the t-plot procedure. The external surface area could then be calculated as the difference between the BET surface area and the micropore surface area. Perhaps this would justify the differences observed for the CeO2-O and –P supports, very similar according to the TEM and XRD results.

Response:

     Thanks for your insightful comment. The CeO2-O displayed H3 type hysteresis loop and verifed the existence of slit-shaped pores and the BET surface area of 67.03m2 g-1, the average pore diameter of 10.69nm and crystallite size of 9.5nm. But the CeO2-P possessed H4 type hysteresis loop at a relatively lower pressure and the BET surface area of 80.06m2 g-1, the average pore diameter of 3.51nm and crystallite size of 6.3 nm. The external surface of two are indeed difference. The detail was in line 97 to 114 on page 4 and Table 1.

6)XPS:Table 2, enclosing the surface atomic concentrations, should be completed with experimentally determined bulk compositions (Fe/Ce, Fe/W), for instance by ICP.

Response:

    Thanks for your good comment. We should be completed with experimentally determined bulk compositions (Fe/Ce, Fe/W), for instance by ICP originally. The NH3-SCR reaction was mainly carried out on the surface of the catalyst. And from Table 2, the surface compositions (Fe/Ce, Fe/W) of all samples were similar. It can draw the right conclusion under known conditions. We could use more technical means as ICP to improve our manuscripts in the further research.

7)UV-Visible diffuse reflectance spectra: according to this section, Fe/W-CeO2-P presents the lowest iron dispersion as compare to the other catalysts. This is in total disagreement with the rest of the discussion and results, and is unexpected due to the small crystal size and high BET surface area of the CeO2-P support.

Response:

     Thanks for your good comment. The manuscript has been carefully revised in order to the unexpected results. Finally, the error has been corrected. Meanwhile, subbands >400nm (centered at around 520 nm) was assigned to large Fe2O3 particles (Hong, W.J. et. Applied Catalysis B: Environmental. 2011, 106, 142-148), which was consistent with HRTEM results. This peak can be related to the degree of dispersion of metal. The stronger intensity absorption band of Fe2O3 was detected suggesting that iron was in highly dispersive state, no large iron oxide particles aggregated in the sample (Liu, F. et. Applied Catalysis B: Environmental. 2010, 96, 408-420; Zhang, X. Catalysts 2017, 7). The corresponding modifications are as follows: line 180 to 183 on Page 8 in the revised manuscript.

8)IR-Py: Although the IR band at 1490 cm-1 is assigned to Lewis-Bronsted acid complex, the samples described here do not present any Bronsted acidity (no band is observed at 1540 cm-1), so the 1490 cm-1 band is due only to Lewis acidity.

Response:

     Thanks for your good comment. ndeed, there was no completely consistent explanation for the peak at around 37250px-1. One comment was that the peak around 1490 cm-1 attributed to both L and B acid sites (Y. Jia, J. Wang, K. Zhang, W. Feng, S. Liu, C. Ding, P. Liu. Microporous Mesoporous Mater. 247 (2017) 103-115; M. Schwidder, M. Santhosh Kumar, U. Bentrup, J. Pérez-Ramírez, A. Brückner, W. Grünert, Microporous Mesoporous Mater. 111 (2008) 124-133). Another comment considers that this peak was attributed to the pyridine species interacting with both the two kinds of acid sites (A. Maijanen, E.G. Derouane, J.B. Nagy. Appl. Surf. Sci. 75 (1994) 204-212; F. Jin, Y.D. Li. Catal. Today 145 (2009) 101-107). Nevertheless, for all of them, the peak around 37250px-1 was not included in the calculation of the amounts of Lewis acid sites and bronsted acid sites.

9)Activity measurements: There is a clear difference in the catalytic behavior of Fe/W-CeO2-P and –O, on the one hand, and Fe/W-CeO2-R and –C on the other. The former two are much more active and their conversion curves overlay in a temperature range of 100 to 250ºC. This clearly indicates a direct correlation between the crystal size of the CeO2 support and the NO conversion, and other issues such as oxygen vacancies, or Fe-Ce and Fe-W interactions do not seem to play a determining role at low temperatures. In my opinion the differences in activity between the CeO2-O and the CeO2-P based catalysts are small and do not justify the activity ranking given by the authors in section 3. This Discussion section should be completely rewritten, and clear arguments should be presented correlating the different properties of the four catalysts with their true catalytic behavior. It is important to isolate as much as possible the effects of the morphology, of the crystal size and of the crystal faces exposed by the four CeO2 supports compared on the iron dispersion, on the interaction of iron with the support, on the Fe-W interaction, on the amount of oxygen vacancies, on the acidity. And all those properties have to be correlated in a coherent way, between them and also with the catalytic properties. The authors present a large amount of information regarding the properties of the catalyst. This information has to be analyzed in a global way, not individually, one technique after the other, as it is in the present manuscript. Once the properties of the catalysts are clearly exposed, it will be easier to find correlations with the catalytic behavior.

Response:

     Thanks for the reviewer’s insightful comment. The Fe/W-CeO2-P catalyst has the highest NO conversion in the temperature range of 100-325°C. We demonstrated the relationship between activity and catalyst properties through various characterizations. First, we found the difference in the morphology of the CeO2 carrier by TEM. The literature showed that the difference in morphology was derived from the selective exposure of different crystal surfaces, and the activity of each crystal face exposed was different. Then we could think about whether the activity of the catalyst was related to it. Then we made other conventional characterization methods to find that the exposed crystals face, the distribution of Fe2O3, and the oxygen vacancies and acid sites have significant effects. The Fe/W-CeO2-P showed the largest oxygen vacancies, the largest number of acid sites, the largest BET surface area and the best dispersibility of Fe2O3. As for the Fe/W-CeO2-O catalyst and the Fe/W-CeO2-P catalyst in the low temperature section, the performance was similar, because the two have similar exposed surfaces but are not identical, and the difference was reflected in the high temperature section. Anyway, we revised the manuscript. We could separate as much as possible the separation of four CeO2 supports for iron dispersion, iron-support interaction, iron-to-W interaction, oxygen vacancies and acidity, and the morphology and crystals of the exposed crystal surfaces of the four CeO2 supports, size and surface effects in the following research. The corresponding revision can be found in line 240 to 241 on page 11 and line 250 to 251 on page 11.

10)Finally, the rates given in the text (lines 207-209) are given in mol NO/g·s and the Arrhenius plot is based on the rates determined per m2 of catalyst. This is confusing and leads to false conclusions: according to this plot, catalyst Fe/W-CeO2-C should present a behavior in the 100-150ºC temperature range closer to the –P and –O catalysts than to the nano-rods based catalyst, which is not true. The plot should be given based on rates determined per gram of catalyst. If helpful for discussion, it can also be given per m2, but only as a second option.

Response:

    Thanks for your comment. When calculating activation energy according to Arenius formula, it was necessary to select a lower conversion rate and a lower temperature. However, the apparent activation energy did not change with the change of temperature. According to the formula (The apparent activation energies (Ea) were calculated by the Arrhenius law (ln (k) = f (1000/T))), no matter what to calculate the change of activation energy, it was only the difference of pre-exponential factor, but the activation energy is not affected.

11)Materials and methods: This sections should be placed before the Results and Discussion section. The description of the catalyst preparation has to be improved. Please name the preparations 1, 2, 3 and 4 according to the nomenclature given in the rest of the manuscript (CeO2-O, -P, -R and –C). Is the impregnation performed by incipient wetness? Please specify. Which is the diameter of the reactor? According to section 4.3, the kinetic experiments were limited to NOx conversions below 17%. However, the conversion for the CeO2-O and –P based catalysts in the 100-150ºC range reaches values close to 40%. Are these kinetic experiments different from the ones presented in figure 8? If so, the activity and selectivity plots should also be included. Moreover, although it is important to limit conversion for accurate kinetic measurements, low conversion levels do not ensure absence of mass transfer (external and internal diffusion) control. This will be related to experimental factors such as reactor diameter, length of the catalytic bed, total flow and catalyst particle size.

Response:

     Thanks for your good question. In this Catalysts journal convention, the section of Materials and methods is left out at the back. The description of the catalyst preparation are intended to make it easier for readers to understand what the morphology of the catalyst is. This is also the name given in many literatures (Lianjun Liu.ChemCatChem 2011, 3, 978-989; Xiaojiang Yao. Ind. Eng. Chem. Res. 2018, 57, 12407-12419). I was sorry for the confusion caused to you. The impregnation was performed by incipient wetness. The kinetic experiments were limited to NOx conversions below 40%. The experimental factors were accurate, such as the fixed-bed reactor diameter was 8 mm. In revised manuscript, the corresponding revision can be found in line 277 on page 11, line 295 to 300 on page 11 and line 309 on page 12.

Reviewer 3 Report

The study of nano-structured materials is interesting, anyway the paper shows major faults. The main problem, in my opinion, is the confusing interpretation of characterization results that often appear contadictory.

(1) English language must be improved throughout the text. I cannot suggest line by line corrections because it will take to long.

(2) One of the main results is the enhanced catalytic activity of nanopolyhedra- based composition, anyway it is only slightly more active than ordinary CeO2-based material as reported in Figure 8. The reason for NO drop conversion down to 60% at 350°C with following increase at higher temperature must be explained.

(3) In the introduction (line 48) ...numerous studies...must be cited!

(4) In line 60 a very first reference to ABCD catalysts is shown. To help the reader understand the catalyst compositions  a reference to the catalytst preparation section sholud be added.

(5) A very important information is missing...the loading of active phase F/W of the main four catalysts. This information is crucial for characterization results evaluation.

(6) in line 66 (110) and (100) seem to be inverted according to the pictures. Line 68 : it is not clear the reason why crystal planes are not detected by HRTEM in ordinary ceria, usually in other works they are similar to nanopolyhedra...

(7) From XRD and BET analysis a high dispersion of Fe and W phases is suggested, but, not indicating the loading this information is not proven by direct dispersion measurement and it is suggested as on of the main characteristics to explain enhanced activity.

(8) Raman bands shifting are interpreted as Feand W incorporation into CeO2 lattice, anyway XRD analysis do not show any peak shift...the interpretation of results seems a little contradictoty.

(9) line 129. The oxygen vacancy concentration order shows that CeO2-P material is the less rich in vacancies and the same trend is observed with the supported materials. This contradicts the main statement in the abstract and also XPS results.

Author Response

The study of nano-structured materials is interesting, anyway the paper shows major faults. The main problem, in my opinion, is the confusing interpretation of characterization results that often appear contadictory.

1)English language must be improved throughout the text. I cannot suggest line by line corrections because it will take to long.

Response:

    Thanks for the reviewer’s comment. The manuscript has been carefully revised in order to improve English writing and reduce grammatical errors in English usage. Moreover, we have looked for experts who are native speakers of English to polish this manuscript. The corresponding revision can be found in line 20 to 22 on page 1, line 26 to 31 on page 1, line 38 to 40 on page 2, line 126 to 128 on page 5, line 181 to 182 on page 8 and line 275 to 278 on page 11.

2)One of the main results is the enhanced catalytic activity of nanopolyhedra- based composition, anyway it is only slightly more active than ordinary CeO2-based material as reported in Figure 8. The reason for NO drop conversion down to 60% at 350°C with following increase at higher temperature must be explained.

Response:

    Thanks for the reviewer’s comment. We have tested morphologies of the catalysts, including the Fe/W-CeO2-O, using HRTEM and XRD. Fe/W-CeO2-O also exposed crystalline planes and was similar to polyhedrons, but it also exposed other crystalline planes without definite crystalline planes. Because the two have similar exposed surfaces but were not identical, and the difference was reflected in the high temperature section. The corresponding modifications were as follows: line 68 to 70 on Page 2 in the revised manuscript.

3)In the introduction (line 48) ...numerous studies...must be cited!

Response:

    Thanks for the reviewer’s comment. We have cited the paper (Montini et al. Chem. Rev. 116, 5987–6041; Damma et al. Industrial & Engineering Chemistry Research 57 (2018) 16615-16621) to improve the manuscript. The corresponding modifications were as follows: line 45 and line 49 on Page 2  in the revised manuscript.

4)In line 60 a very first reference to ABCD catalysts is shown. To help the reader understand the catalyst compositions  a reference to the catalytst preparation section sholud be added.

Response:

    Thanks for the reviewer’s comment. We have cited the paper (Hao-Xin Mai. J. Phys. Chem. B 2005, 109, 24380-24385). The corresponding revision can be found in line 62 on page 2.

5)A very important information is missing...the loading of active phase F/W of the main four catalysts. This information is crucial for characterization results evaluation.

Response:

    Thanks for the reviewer’s comment. We were sorry for missing that the loading of active phase F/W of the main four catalysts is 9.5 to 1.0 of molar ratio. The corresponding revision can be found in line 275 to 276 on page 11.

6)in line 66 (110) and (100) seem to be inverted according to the pictures. Line 68 : it is not clear the reason why crystal planes are not detected by HRTEM in ordinary ceria, usually in other works they are similar to nanopolyhedra.

Response:

     Thanks for the reviewer’s comment. We tested all morphologies of the catalysts, including the Fe/W-CeO2-O, using HRTEM and XRD. Fe/W-CeO2-O also exposed crystalline planes and was similar to polyhedrons, but it also exposed other crystalline planes without definite crystalline planes. Thank you again for your comment. The corresponding modifications were as follows: line 68 to 70 on Page 2 in the revised manuscript.

7)From XRD and BET analysis a high dispersion of Fe and W phases is suggested, but, not indicating the loading this information is not proven by direct dispersion measurement and it is suggested as on of the main characteristics to explain enhanced activity.

Response:

    Thanks for your insight comment. To test the distribution of elements in catalysts, EDS color mapping should be done. This was a good suggestion. We were sorry not to do it, for laboratory conditions were not allowed to do it. In the following, we could create conditions to explore EDS color mapping.

8)Raman bands shifting are interpreted as Fe and W incorporation into CeO2 lattice, anyway XRD analysis do not show any peak shift...the interpretation of results seems a little contradictoty.

Response:

    Thanks for your comment. In fact, if you look closely at XRD, you can see the difference in peak intensity and peak width of the four samples. The above represents the crystallinity and particle size of the sample. The corresponding modifications were as follows: line 87 to 89 on Page 3 in the revised manuscript.

9) line 129. The oxygen vacancy concentration order shows that CeO2-P material is the less rich in vacancies and the same trend is observed with the supported materials. This contradicts the main statement in the abstract and also XPS results.

Response:

    Thank you for your comment. Oxygen vacancies played an important role in the NH3-SCR process. As you can see, Raman spectroscopy, because of its weak absorption, gave information from both the volume and surface of the sample. XPS provided information from the surface of the sample. The NH3-SCR reaction occurred on the surface of the catalyst, so we believed that this reaction was more closely related to the oxygen vacancies on the catalyst surface. In summary, the trend of catalyst activity was consistent with the trend of surface oxygen vacancies, and we had more reasons to believe this. The corresponding revision can be found in line 249 to 251 on Page 11 in the revised manuscript.

Round 2

Reviewer 1 Report

Accept in its present form.

Author Response

Thank you

Reviewer 2 Report

General comments.

The authors have addresses some of the suggestions made and the manuscript has been slightly improved as compared to the first version. Thus, English usage has been revised at some points of the manuscript, and additional references have been included. However, the main discrepancies regarding oxygen vacancies and iron dispersion as determined by different characterization techniques and the correlation of these catalyst properties with the activity are not clarified.

I am sorry, but in my opinion, the manuscript is still far from suitable for publication in Catalysts.

More details are given below.

Specific comments.

1.      Page 2, line 60: As mentioned in the previous revision, the ordinary CeO2 cannot be defined as an amorphous material. It presents the typical XRD patter of the fluorite cubic structure and, according to the authors, “Fe/W-CeO2-O also exposed crystalline planes and was similar to polyhedrons, but it also exposed other crystalline planes without definite crystalline planes”. The differences between this material (CeO2-O) and the nanopolyhedrons (CeO2-P) should be clearly exposed at the beginning of the manuscript.

2.      TEM and HRTEM: in the case of the nanorods and the cubes the Fe2O3 and FeWO4 particles can be distinguished from the CeO2 support because of the different morphology. However, in the case of the ordinary CeO2 and the nanopolyhedra the authors make the distinction based on the interplanar distance, which in some cases is very similar (0.29 nm for  (111) crystal plane of FeWO4 and 0.31 nm for the (111) crystal plane of CeO2). In these two cases, it would be convenient to determine the chemical composition, for instance by elastic collision (STEM) or inelastic collision based techniques (EDX, EELS).

Response:      Thanks for your insightful comment. The ordinary CeO2 also exposed crystalline planes and was similar to polyhedrons, but it also exposed other crystalline planes without definite crystalline planes. It could make the distinction based on the interplanar distance, which can distinguish the ordinary CeO2 and the nanopolyhedra. Of course, the method provided about elastic collision (STEM) or inelastic collision based techniques (EDX, EELS) can better distinguish the two. We can achieve the same goal with the previous revision. The corresponding modifications were as follows: line 68 to 70 on Page 2 in the revised manuscript.

What I mend by this comment was that it was not possible, just by looking at the pictures and spacings, to distinguish the FeWO4 from the CeO2 support in the case of the nanopolyhedra CeO2. I am sorry I did not express myself clearly. In any case, I still believe it is difficult to differentiate the FeWO4 from the support in the case of the ordinary and nanopolyhedra CeO2, only by measuring the interplanar spacing.

3.      BET: as the authors mention in lines 103-104, the porosity observed is probably due to the presence of intercrystalline voids between the nanocrystallites. If some microporosity is formed as a consequence of the interaction of iron and cerium, it could be determined by means of the t-plot procedure. The external surface area could then be calculated as the difference between the BET surface area and the micropore surface area. Perhaps this would justify the differences observed for the CeO2-O and –P supports, very similar according to the TEM and XRD results.

Response:      Thanks for your insightful comment. The CeO2-O displayed H3 type hysteresis loop and verifed the existence of slit-shaped pores and the BET surface area of 67.03m2 g-1, the average pore diameter of 10.69nm and crystallite size of 9.5nm. But the CeO2-P possessed H4 type hysteresis loop at a relatively lower pressure and the BET surface area of 80.06m2 g-1, the average pore diameter of 3.51nm and crystallite size of 6.3 nm. The external surface of two are indeed difference. The detail was in line 97 to 114 on page 4 and Table 1.

I am afraid that also in this case I have not explained myself properly. The values that the authors have included in Table 1 are clear. And the hysteresis loop presented by Fe/W-CeO2-P resembles a H4 type hysteresis , which is, indeed, found in the case of solids presenting a dual micro-mesoporosity, such as nanocrystalline or mesoporous zeolites and micro-mesoporous carbons. My suggestion is to include the micropore volume or micropore surface area, determined by the t-plot procedure, in Table 1 for all the catalysts, in order to proof the generation of microporosity in the case of the CeO2-P based catalyst. Moreover, in their response, the authors mention differences in the external surface area. However, these values are not shown in the revised manuscript.

4.      Regarding the amount of oxygen vacancies, the order given by Raman spectroscopy (page 5, lines 130-134, R>C>P>O) is different than the one obtained from the XPS data (page 6, lines 144-145, P>O>R>C). Please revise the data or add an explanation for this disagreement. If crystal plane (111) is the most favored in the CeO2-P support, and this plane is the most stable according to the literature, how do the authors explain a larger amount of oxygen vacancies? Moreover, if Fe2+ is formed by a redox reaction of Fe3+ with CeO2, and the reduction of Fe3+ concentration follows the order P>O>C>R, Fe/W-CeO2-P should present the lowest Fe3+ concentration or the highest Fe2+ concentration. Please revise lines 154-158 in pages 6 and 7.

5.      UV-Visible diffuse reflectance spectra: despite the mofidications made by the authors, Fe/W-CeO2-P presents the lowest iron dispersion as compared to the other catalysts (see line 183, page 8). This is in total disagreement with the rest of the discussion and results, and is unexpected due to the small crystal size and high BET surface area of the CeO2-P support. This issue is, in my opinion, not clarified.

6.      Activity measurements: There is a clear difference in the catalytic behavior of Fe/W-CeO2-P and –O, on the one hand, and Fe/W-CeO2-R and –C on the other. The former two are much more active and their conversion curves overlay in a temperature range of 100 to 250ºC, although the CeO2-P based catalyst presents higher activity in the 250-300ºC range. I only see a clear correlation with crystal size and/or BET surface area, and very small differences when comparing the CeO2-P and CeO2-O based catalysts. Thus, crystal planes exposed do not play a determinant role, as the CeO2-O presents more than one crystal plane exposed and not a preferred one.

If still we consider the CeO2-P as the most active catalyst (although only at high temperatures) this higher activity can only be correlated with its higher BET surface area and higher Lewis acid site density, two properties that could be direct consequence of its smaller crystal size. Regarding oxygen vacancies, CeO2-P presents the highest amount according to XPS or less amount than the CeO2-R and –C supports according to Raman, and regarding iron dispersion, the discussion of the UV-Visible diffuse reflectance spectra is not clear.

7.      The discussion section has not been rewritten. No arguments trying to explain the differences observed by Raman and XPS or the unexpected low iron dispersion deduced from the UV-Vis results. A catalyst activity ranking is given, which can be directly deduced from the activity section, and a new sentence has been included: “However, Raman results showed that the total oxygen vacancies in Fe/W-CeO2-P catalysts was not the most, which further indicated that the surface oxygen vacancies form the exposed CeO2 (111) crystal plane played a significant role in NH3-SCR.” This sentence does not contribute to the discussion with any valid argument. It is a mere speculation.   

Catalyst Fe/W-CeO2-P is not mcuh better than the CeO2-O based catalyst, the conclusions regarding this catalyst do not correspond to the results presented and the effect of the exposure of different CeO2 crystal surfaces on the catalytic activity is not clearly demonstrated. The paper has to be improved substantially before being suitable for publication in Catalyst.

Author Response

The authors have addresses some of the suggestions made and the manuscript has been slightly improved as compared to the first version. Thus, English usage has been revised at some points of the manuscript, and additional references have been included. However, the main discrepancies regarding oxygen vacancies and iron dispersion as determined by different characterization techniques and the correlation of these catalyst properties with the activity are not clarified. I am sorry, but in my opinion, the manuscript is still far from suitable for publication in Catalysts. More details are given below. Specific comments. 1. Page 2, line 60: As mentioned in the previous revision, the ordinary CeO2 cannot be defined as an amorphous material. It presents the typical XRD patter of the fluorite cubic structure and, according to the authors, “Fe/W-CeO2-O also exposed crystalline planes and was similar to polyhedrons, but it also exposed other crystalline planes without definite crystalline planes”. The differences between this material (CeO2-O) and the nanopolyhedrons (CeO2-P) should be clearly exposed at the beginning of the manuscript. Response: Thanks for your good comment. We are very sorry that we did not understand your comment thoroughly in the previous revision. CeO2-O is from a direct calcination of cerium nitrate, the details in line 272 to 273 on page 12. From TEM and HRTEM, CeO2-O morphology include an ellipsoid and a polyhedron, and the exposed crystal plane containing CeO2 (111), (110) and (100). CeO2-P was prepared  by different methods of CeO2-O, the details in line 263 to 266 on page 12. CeO2-P morphology is just  polyhedron. We  revise the manuscript. The corresponding modifications were as follows: line 62 to 63 on Page 2 and line 69 to 70 on Page 2 in the revised manuscript. 2. TEM and HRTEM: in the case of the nanorods and the cubes the Fe2O3 and FeWO4 particles can be distinguished from the CeO2 support because of the different morphology. However, in the case of the ordinary CeO2 and the nanopolyhedra the authors make the distinction based on the interplanar distance, which in some cases is very similar (0.29 nm for  (111) crystal plane of FeWO4 and 0.31 nm for the (111) crystal plane of CeO2). In these two cases, it would be convenient to determine the chemical composition, for instance by elastic collision (STEM) or inelastic collision based techniques (EDX, EELS). Response: Thanks for your insightful comment. The ordinary CeO2 also exposed crystalline planes and was similar to polyhedrons, but it also exposed other crystalline planes without definite crystalline planes. It could make the distinction based on the interplanar distance, which can distinguish the ordinary CeO2 and the nanopolyhedra. Of course, the method provided about elastic collision (STEM) or inelastic collision based techniques (EDX, EELS) can better distinguish the two. We can achieve the same goal with the previous revision. The corresponding modifications were as follows: line 68 to 70 on Page 2 in the revised manuscript. What I mend by this comment was that it was not possible, just by looking at the pictures and spacings, to distinguish the FeWO4 from the CeO2 support in the case of the nanopolyhedra CeO2. I am sorry I did not express myself clearly. In any case, I still believe it is difficult to differentiate the FeWO4 from the support in the case of the ordinary and nanopolyhedra CeO2, only by measuring the interplanar spacing. Response: Thanks for your detailed comment. It is all our fault. We have repeat the characterization many times, and the results are the same each time. So we have the confidence to believe this conclusion.  From the literature (Hui Wang, Zhenping Qu, et. Environ. Sci. Technol. 2016, 50, 13511-13519; Zhang, J, Zhang, Y. et. J. Nanopart. Res. 2012, 14 (4), 1-10) we can  arrive at this conclusion form XRD. The corresponding modifications were as follows: line 91 to 95 on Page 3 in the revised manuscript. 3.      BET: as the authors mention in lines 103-104, the porosity observed is probably due to the presence of intercrystalline voids between the nanocrystallites. If some microporosity is formed as a consequence of the interaction of iron and cerium, it could be determined by means of the t-plot procedure. The external surface area could then be calculated as the difference between the BET surface area and the micropore surface area. Perhaps this would justify the differences observed for the CeO2-O and –P supports, very similar according to the TEM and XRD results. Response: Thanks for your insightful comment. The CeO2-O displayed H3 type hysteresis loop and verifed the existence of slit-shaped pores and the BET surface area of 67.03m2 g-1, the average pore diameter of 10.69nm and crystallite size of 9.5nm. But the CeO2-P possessed H4 type hysteresis loop at a relatively lower pressure and the BET surface area of 80.06m2 g-1, the average pore diameter of 3.51nm and crystallite size of 6.3 nm. The external surface of two are indeed difference. The detail was in line 97 to 114 on page 4 and Table 1. I am afraid that also in this case I have not explained myself properly. The values that the authors have included in Table 1 are clear. And the hysteresis loop presented by Fe/W-CeO2-P resembles a H4 type hysteresis , which is, indeed, found in the case of solids presenting a dual micro-mesoporosity, such as nanocrystalline or mesoporous zeolites and micro-mesoporous carbons. My suggestion is to include the micropore volume or micropore surface area, determined by the t-plot procedure, in Table 1 for all the catalysts, in order to proof the generation of microporosity in the case of the CeO2-P based catalyst. Moreover, in their response, the authors mention differences in the external surface area. However, these values are not shown in the revised manuscript. Response: Thanks for your detailed and patient comment. We are really sorry for not having correctly understood the previous recommendations. We will carefully correct and add the necessary content. The micropore surface area have included in Table 1. We mention that differences in the external surface area in our response is a mistake in writing. And it should be differences in the BET surface area or micropore surface area. The corresponding revision can be found in line 101, 106 and line 117 to 119 on page 4. 4.      Regarding the amount of oxygen vacancies, the order given by Raman spectroscopy (page 5, lines 130-134, R>C>P>O) is different than the one obtained from the XPS data (page 6, lines 144-145, P>O>R>C). Please revise the data or add an explanation for this disagreement. If crystal plane (111) is the most favored in the CeO2-P support, and this plane is the most stable according to the literature, how do the authors explain a larger amount of oxygen vacancies? Moreover, if Fe2+ is formed by a redox reaction of Fe3+ with CeO2, and the reduction of Fe3+ concentration follows the order P>O>C>R, Fe/W-CeO2-P should present the lowest Fe3+ concentration or the highest Fe2+ concentration. Please revise lines 154-158 in pages 6 and 7. Response: Thanks for your detailed and patient comment. We think that the comments about oxygen vacancies and Fe3+ may make sense. So we have recalculated the oxygen vacancies according to Raman spectroscopy. We revised the unreasonable error. And we re-fit the curve carefully. And found that your comments is recommendable. Thank you very much for your comments to make the manuscript more reasonable. We have revised the manuscript. The corresponding revision can be found in line 138 to 139 and 141 on page 5, line 143 on page 6 of Figure 4c, line 153 to 154 on pages 6, line 162 to 165 on pages 6 and 7 and line 179 on page 7 of Table 2. 5.      UV-Visible diffuse reflectance spectra: despite the mofidications made by the authors, Fe/W-CeO2-P presents the lowest iron dispersion as compared to the other catalysts (see line 183, page 8). This is in total disagreement with the rest of the discussion and results, and is unexpected due to the small crystal size and high BET surface area of the CeO2-P support. This issue is, in my opinion, not clarified. Response: Thanks for your good comment. We tested the UV-visible spectrum carefully again, and the results were revised in the manuscript. The corresponding revision can be found in line 182 to 194 on page 8 including Figure 6. 6.  Activity measurements: There is a clear difference in the catalytic behavior of Fe/W-CeO2-P and –O, on the one hand, and Fe/W-CeO2-R and –C on the other. The former two are much more active and their conversion curves overlay in a temperature range of 100 to 250ºC, although the CeO2-P based catalyst presents higher activity in the 250-300ºC range. I only see a clear correlation with crystal size and/or BET surface area, and very small differences when comparing the CeO2-P and CeO2-O based catalysts. Thus, crystal planes exposed do not play a determinant role, as the CeO2-O presents more than one crystal plane exposed and not a preferred one. If still we consider the CeO2-P as the most active catalyst (although only at high temperatures) this higher activity can only be correlated with its higher BET surface area and higher Lewis acid site density, two properties that could be direct consequence of its smaller crystal size. Regarding oxygen vacancies, CeO2-P presents the highest amount according to XPS or less amount than the CeO2-R and –C supports according to Raman, and regarding iron dispersion, the discussion of the UV-Visible diffuse reflectance spectra is not clear. Response: Thanks for your detailed and patient comment. Both CeO2-P and CeO2-O have too many similarities in terms of morphology and exposed crystal faces. Therefore, in the subsequent oxygen vacancies, the BET specific surface area, the dispersion of iron oxide, and the amount of acid all have close performance, which is also reflected in the catalyst activity in the low temperature section. As the revisions stated, the crystal faces may not be the main influencing factors of both. However, the difference in the preparation methods (Li, B.; Wang, H. etc. Acta Physico-Chimica Sinica 2013, 29, 1289-1296; Fang, D. etc. Fuel Processing Technology 2015, 134, 465-472; He, L. F. Chemical Journal of Chinese Universities-Chinese 2012, 33, 2532-2536.) may be the main influencing factors, and the incomplete agreement between the two exposed crystal faces leads to a significant difference in the catalytic activity in the high temperature range. We repeated a lot of tests and calculations and improved the manuscript. The corresponding changes are reflected in the above responses. 7. The discussion section has not been rewritten. No arguments trying to explain the differences observed by Raman and XPS or the unexpected low iron dispersion deduced from the UV-Vis results. A catalyst activity ranking is given, which can be directly deduced from the activity section, and a new sentence has been included: “However, Raman results showed that the total oxygen vacancies in Fe/W-CeO2-P catalysts was not the most, which further indicated that the surface oxygen vacancies form the exposed CeO2 (111) crystal plane played a significant role in NH3-SCR.” This sentence does not contribute to the discussion with any valid argument. It is a mere speculation.   Catalyst Fe/W-CeO2-P is not mcuh better than the CeO2-O based catalyst, the conclusions regarding this catalyst do not correspond to the results presented and the effect of the exposure of different CeO2 crystal surfaces on the catalytic activity is not clearly demonstrated. The paper has to be improved substantially before being suitable for publication in Catalyst. Response: Thanks for your detailed and heartfelt comment. We reanalyzed the Raman data and re-test UV/vis, arriving at a suitable conclusion. This is explained in Response 4 and 5. The discussion section has been rewritten. The corresponding revision can be found in line 248 to 281 on page 11 and 12.

Reviewer 3 Report

The paper has been considerably improved and I suggest it for publication in Catalysts.

Author Response

Thank you 

Round 3

Reviewer 2 Report

General comments.

In this version, the authors have addressed all the suggestions made by the Referees. The results presented in the manuscript are now coherent and they all lead to the same conclusions. English usage should be revised at the points where new text has been added and two minor modifications should be made, but the manuscript is now ready for publication in Catalysts.

Specific comments.

1.      Page 4, Table 1. The table has been completed with the values of micropore surface area. However, this value is higher than the BET surface area for the four catalysts, and this is not possible. Probably the authors should recalculate the micropore surface adjusting the t-plot in a different range of thickness of the adsorbant layer. Depending on the textural properties of the sample the t-plot can be adjusted in the range of 5.5-8.0 Å (microporous solids, such as zeolites) or in the range of 3.5-5.0 Å (mesoporous materials such as delaminated zeolites, MCM-41…). Micropore surface area has to be equal or lower than BET. The difference between these two values corresponds to the external surface area.

2.      Page 6, lines 154-156:  “This indicated that the oxygen vacancies of the Fe/W-CeO2-P was more than those of Fe/W-CeO2-O, Fe/W-CeO2-R and Fe/W-CeO2-C, which is in agreement with the result of the oxygen vacancies by XPS.” I think the authors are referring to Raman and not to XPS in this sentence.

Author Response

1. Page 4, Table 1. The table has been completed with the values of micropore surface area. However, this value is higher than the BET surface area for the four catalysts, and this is not possible. Probably the authors should recalculate the micropore surface adjusting the t-plot in a different range of thickness of the adsorbant layer. Depending on the textural properties of the sample the t-plot can be adjusted in the range of 5.5-8.0 Å (microporous solids, such as zeolites) or in the range of 3.5-5.0 Å (mesoporous materials such as delaminated zeolites, MCM-41…). Micropore surface area has to be equal or lower than BET. The difference between these two values corresponds to the external surface area.

Response:

Thanks for your detailed and patient comment. We are sorry for the error. We have recalculate the micropore surface adjusting the t-plot in a different range of thickness of the adsorbant layer. We We give the t-plot Micropore Area data in Manuscript. The corresponding revision can be found in Table 1  and line 108 on page 4.

2. Page 6, lines 154-156:  “This indicated that the oxygen vacancies of the Fe/W-CeO2-P was more than those of Fe/W-CeO2-O, Fe/W-CeO2-R and Fe/W-CeO2-C, which is in agreement with the result of the oxygen vacancies by XPS.” I think the authors are referring to Raman and not to XPS in this sentence.

Response:

Thanks for your good comment. We carefully reviewed the manuscript again. We think that you have correct opinion aboult that sentence. We have amended the sentence. The corresponding revision can be found in line 154 to 156 on page 6.